# Isolate Whey Protein Promotes Fluid Balance and Endurance Capacity Better Than Isolate Casein and Carbohydrate-Electrolyte Solution in a Warm, Humid Environment

**DOI:** 10.3390/nu15204374

**Published:** 2023-10-16

**Authors:** Mahdi Gholizadeh, Abolfazl Shakibaee, Reza Bagheri, Donny M. Camera, Hossein Shirvani, Frederic Dutheil

**Affiliations:** 1Exercise Physiology Research Center, Life Style Institute, Baqiyatallah University of Medical Sciences, Tehran 1435916471, Iran; m.gholi66@gmail.com (M.G.); shirvani@bmsu.ac.ir (H.S.); 2Department of Exercise Physiology, University of Isfahan, Isfahan 81746-73441, Iran; reza.bagheri@alumni.um.ac.ir; 3Department of Health and Biostatistics, Swinburne University, Melbourne, VIC 3122, Australia; dcamera@swin.edu.au; 4Physiological and Psychosocial Stress, CNRS UMR 6024, LaPSCo, University Clermont Auvergne, Witty Fit, 63000 Clermont-Ferrand, France; fred_dutheil@yahoo.fr; 5Preventive and Occupational Medicine, University Hospital of Clermont-Ferrand (CHU), 63000 Clermont-Ferrand, France

**Keywords:** rehydration, fluid retention, net fluid balance

## Abstract

Protein ingestion is known to enhance post-exercise hydration. Whether the type of protein (i.e., whey, casein) can alter this response is unknown. Accordingly, this study aimed to compare the effects of the addition of milk-derived whey isolate or casein protein to carbohydrate-electrolyte (CE) drinks on post-exercise rehydration and endurance capacity. Thirty male soldiers (age: 24 ± 2.1 y; VO_2max_: 49.3 ± 4.7 mL/kg/min) were recruited. Upon losing ~2.2% of body mass by running in warm and humid conditions (32.3 °C, 76% relative humidity [RH]), participants ingested either a CE solution (66 g/L carbohydrate [CHO]), or CE plus isolate whey protein (CEW, 44 g/L CHO, 22 g/L isolate whey), or CE plus isolate casein protein (CEC, 44 g/L CHO, 22 g/L isolate casein) beverage in a volume equal to 150% of body mass loss. At the end of the 3 h rehydration period, a positive fluid balance was higher with CEW (0.22 L) compared to CEC (0.19 L) and CE (0.12 L). Overall mean fluid retention was higher in CEW (80.35%) compared with the CE (76.67%) and CEC trials (78.65%). The time of the endurance capacity test [Cooper 2.4 km (1.5 miles) run test] was significantly higher in CEC (14.25 ± 1.58 min) and CE [(12.90 ± 1.01 min; (*p* = 0.035)] than in CEW [(11.40 ± 1.41 min); (*p* = 0.001)]. The findings of this study indicate that the inclusion of isolate whey protein in a CE solution yields superior outcomes in terms of rehydration and enhanced endurance capacity, as compared to consuming the CE solution alone or in conjunction with isolate casein protein.

## 1. Introduction 

Tactical athletes, such as military personnel, sometimes face the need to perform physically demanding tasks in circumstances characterized by high levels of heat stress. The concomitant loss of bodily water has implications for cardiovascular and thermoregulatory functioning, perhaps leading to a decrease in work capacity [1]. One of the primary challenges faced by tactical athletes and soldiers while engaging in prolonged physical activity in hot and humid conditions is the depletion of bodily fluids, leading to a reduction in blood volume. This reduction in blood volume has implications for thermoregulation and poses dangers to both physical and cognitive performance [2]. Consequently, the replenishment of bodily fluids lost during physical activity in high temperatures has significance in maintaining optimal performance levels and mitigating the risk of injury. Nevertheless, the complexity of fluid replacement recommendations arises from a dearth of understanding about the optimal time, quantity, and content of drinks to be ingested. Given the importance of physical capacity in military operations, it is essential for planners to possess knowledge about the mitigation of heat-stress-induced performance decline and heat-related injuries via the use of tailored beverages. Furthermore, in cases when the allotted recovery time is constrained to a duration of less than 12 h, it may be imperative to use efficacious hydration measures in order to facilitate expeditious recovery after physical exertion [3,4,5].

It is recommended that following dehydrating exercise, athletes training and competing for sports consume fluids equal to 150% of fluid lost during exercise [6]. Furthermore, it is advised that the hydration solution used should be adequate in facilitating the restoration of electrolyte balance and promoting fluid retention [7,8,9,10,11,12]. Multiple studies have provided evidence supporting the notion that, beyond electrolytes, low-fat milk offers more advantages for replenishing fluids compared to traditional carbohydrate (CHO)-electrolyte (CE) solutions [13,14,15]. Milk inherently has comparable quantities of CHO and electrolytes to CE solutions. Also, protein as a distinct constituent in milk, may potentially have supplementary influences on fluid retention. Indeed, multiple studies have shown that the consumption of protein-containing solutions after exercise yields superior rehydration outcomes compared to CE solutions. In a study conducted by Seifert et al. (2006), it was shown that the inclusion of milk protein in a conventional sports beverage resulted in enhanced water retention [16]. However, it is important to note that the participants consumed an amount equivalent to their decreased body mass, and the nutritional composition of the drinks did not match in terms of calorie and electrolyte content [16]. James et al. (2013) observed that the addition of milk protein to a CE solution yielded greater efficacy compared to a CE solution with equivalent calorie and electrolyte content [17]. Also, Li et al. (2015) reported that protein-CHO solution ingestion after an hour of running at 65% VO_2max_ at normal temperature (24 °C) caused effective rehydration during the 4 h recovery period [18]. Moreover, Li et al. (2018) suggested that consuming 22 g of whey protein in combination with a CE solution during a 4 h recovery period after 60 min of treadmill running with an intensity of 65% VO_2max_ at normal temperature (24 °C) accelerated the rehydration process compared to consuming 15 g of casein protein [19]. In addition, ingestion of 20 g of whey protein together with a CE beverage containing 60 g of CHO/L over a 4 h recovery period at a temperature of 35°C and relative humidity of 60% resulted in a considerable elevation in plasma volume, glucose, and albumin levels [20]. Nevertheless, the precise potential impacts of milk protein could not be ascertained due to the lack of compositional equivalence in prior research including milk solutions. Hence, the objective of this investigation was to assess the effects of whey isolate or casein on the restoration of body fluid-electrolyte equilibrium, regulation of body temperature, and subsequent physical performance during exercise in warm and humid conditions among a group of young male soldiers.

## 2. Methods

### 2.1. Participants 

A total of 30 male soldiers from an Iranian military university (age: 24 ± 2.1 y; VO_2max_: 49.3 ± 4.7 mL/kg/min) voluntarily consented to take part in the research. Participants were randomly assigned to three groups of CHO-electrolyte (CE; n = 10; 66 g/L CHO), CE plus isolate whey protein (CEW; n = 10; 44 g/L CHO, 22 g/L isolate whey), or CE plus isolate casein protein (CEC; n = 10; 44 g/L CHO, 22 g/L isolate casein). Participants were randomly assigned to one of three groups using an online resource (www.randomizer.org, accessed on 13 October 2023). During the period of investigation, all of the participants were selected from a singular platoon, exhibiting comparable daily routines and fitness levels, and adhering to a uniform dietary regimen. The participants were required to submit a medical history form before their involvement in the study. Following a comprehensive clarification of the trial’s information and protocols, all participants provided their written permission by signing a formal declaration. The study received approval from the Clinical Research Ethical Committee of The Baqiyatayyllah University of Medical Science (IR.BMSU.BAQ.REC.1400.044), Tehran, Iran.

### 2.2. Preliminary Testing

Participants completed preliminary testing including body composition, anthropometry, and an endurance capacity test. During the preliminary test, participants underwent a field running test (2400 m) to determine the time to exhaustion and maximum oxygen uptake (VO_2max_) from the regression equation (483/time + 3.5) [21]. Body fat percentage was estimated from a 3-point skin-fold analysis (chest, abdomen, and thigh) using a Harpenden caliper. All measurements were taken on the right side of the body. After measuring the thickness of subcutaneous fat, the percentage of fat was calculated using the Jackson-Pollock formula [22].

### 2.3. Experimental Protocol

The experimental design was a randomized controlled trial. A protocol timeline is outlined in Figure 1. 

The protocol of the present study includes four main steps. The participants were given instructions to document specific information on their food consumption and physical activity. They were also advised to refrain from engaging in any intense physical exercise within 24 h leading up to the day of the experiment. Additionally, they were directed to drink 500 mL of plain water in the evening before their bedtime. At 90 min before the start of the trial, following overnight fasting (10–12 h), to ensure euhydration participants ingested 500 mL of plain water and consumed a standard pre-test breakfast (08:00 a.m.). Next, the participants conducted a bout of typical military exercise to induce fatigue and dehydration until loss of ≥2% of their pre-exercise body mass, considered minimally effective dehydration to impact exercise capacity [2]. Ratings of perceived exertion (RPE) and thermal sensations were obtained using the Borg Scale [23] and a 21-point linear scale ranging from unbearable cold (−10) to unbearable heat (+10). Skin temperatures (T12L, Digital Thermometer, Guangdong, China) were recorded at 5 min intervals throughout the exercise. No fluid was ingested during the dehydrating exercise. Net fluid balance (NFB) was calculated relative to the baseline time point, taking into account the volumes of fluid lost through sweat during exercise (estimated from total body mass loss during exercise), beverage ingested and cumulative urine produced. Participants were in positive fluid balance if the obtained value was >0, and in negative fluid balance if this value was <0. 

Following exercise, the effect of treatment on rehydration was determined by ingestion of one of the three solutions matched for energy density and electrolyte content (Table 1): (a), CHO and electrolytes, sodium, and potassium (CE); (b) CHO, isolate whey protein (Suppland Nutrition Isolate Whey Protein Powder, Tehran, Iran), and the same electrolytes as CE (CEW); and (c) CHO, isolate casein protein (Suppland Nutrition Isolate Casein Protein Powder, Tehran, Iran), and the same electrolytes as CE (CEC). The solutions were mixed for 1 h before ingestion and the temperature of the drink at serving was 18.2 °C. Total urine volume and blood samples were collected pre- and post-3 h rehydration period, in which participants were asked to empty their bladder as much as possible and followed by a measurement of nude body mass. In addition to the physiological measures, participants were asked to perceive thirst, abdominal discomfort, and stomach fullness, at the end of each hour during recovery. The answers of the participants were scaled from 0 to 10, in which 0 meant “not so much” and 10 meant “very much”. After 3 h of rehydration, participants completed the field-expedient options to measure endurance capacity test. The test consists of field running (2400 m). The test time was recorded in minutes and seconds with a stopwatch.

### 2.4. Laboratory Analysis

All blood and urine samples were measured immediately after collection in duplicate. Plasma glucose concentration was analyzed using a glucose kit enzymatic calorimetric method (Pars Azmun, Tehran, Iran). Hematocrit was determined by microcentrifuge (M.L 25r.Parsazmaco, Tehran, Iran). Hemoglobin concentration was measured by the cyanmethemoglobin method. The percentage change in plasma volume (PV) was calculated based on hemoglobin and hematocrit values [24]. Urine specific gravity (USG) was measured using a USG analyzer (PEN-Urine S. G., Tokyo, Japan). Serum and urine samples were analyzed for sodium and potassium concentrations by flame photometry (BWB XP, London, UK). 

### 2.5. Statistical Analysis

Data are presented as mean ± standard deviation (SD). All statistical calculations were performed using SPSS Version 22. Data were checked for normality of distribution using a Shapiro–Wilk test. A two-way (trial × time) repeated measures ANOVA with Bonferroni post-hoc test was used to determine differences between time points and treatments. Variables containing one factor were analyzed using a one-way ANOVA. Bonferroni-adjusted paired-sample student’s t-test was used to compare the means and locate significant differences for the three pairwise contrasts, accordingly. The significance for the adjusted contrasts was accepted at *p* ≤ 0.01. Effect size (ES) was also expressed as the standardized difference and 95% confidence intervals (CI) were calculated for all dependent variables using the Hedge’s g method corrected for bias. ES was interpreted as trivial, small, medium, and large for values 0.00–0.20, <0.50, <0.8, and ≥0.8, respectively. 

## 3. Results

Table 2 indicates the baseline characteristics of participants included in the present study. There was no significant difference between groups for any variable at baseline (*p* > 0.05). At the end of the study, three participants from CE and two from other groups dropped out due to some personal issues. 

The effect of the dehydrating exercise on primary outcome parameters is shown in Table 3. Participants lost on average 2.1 ± 0.2% of their initial body mass. The average time to dehydration was 12.7 min ± 1.7 (*p* = 0.418). During the rehydration period, the mean volume of fluid consumed was 2235 ± 188 mL (CE), 2287 ± 203 mL (CEW), and 2303 ± 246 mL (CEC) (F = 0.2, *p* = 0.820).

At the baseline, there were no differences observed between trials in the urine volume, USG, and blood glucose (*p* > 0.05). The total volume of urine produced between trials (CE: 518 ± 19 mL; CEW: 440 ± 35 mL; CEC: 492 ± 31 mL; *p* = 0.037), meaning that in the CEW trial ingested fluid retention was approximately 80.35% of the ingested solution by the end of the 3 h rehydration period (Phase 3). This value was greater than that of the CE and CEC trials (CE vs. CW vs. CEC: 76.67 ± 2.8% vs. 80.35 ± 5.9% vs. 78.65 ± 2.6%; F = 1.48, *p* = 0.251). 

### 3.1. Electrolyte Balance

Electrolyte changes are shown in Figure 2. There was no significant main effect between trials in urine and serum sodium levels (*p* = 0.294) or interaction effect of test drink over time (*p* = 0.301) during rehydration. During the rehydration phase, the sodium returned to the euhydrated levels in all trials. There were no significant differences in the urine potassium levels between trials at any time (*p* = 0.573). There was a main effect of time between trials on serum potassium levels (*p* = 0.042) during rehydration and an interaction effect of test solution over time (*p* = 0.031). Serum potassium levels in the CEW trial were significantly higher during rehydration compared with the CE trial (*p* = 0.025) and CEC trial (*p* = 0.018).

### 3.2. NFB and Fluid Retention

Mean NFB after 3 h rehydration was higher in the CEW trial (0.22 L) compared with the CE [(0.12 L); (*p* = 0.001)] and CEC trial [(0.19 L); (*p* = 0.031)]. No significant difference was found between CEW and CEC trials (*p* = 0.061). Overall mean fluid retention was higher in the CEW trial (80.35 ± 5.9%) compared with the CE (76.67 ± 2.8%) (*p* = 0.301) and CEC trial [(78.65 ± 2.6%); (*p* = 0.816)]. No significant difference was found between trials (*p* > 0.05).

The percent change in plasma volume (PV) from the baseline levels is shown in Figure 3. Time had a significant effect (*p* = 0.001). Compared to pre-exercise, PV decreased post-dehydration during all trials (*p* = 0.001) and increased at 3 h during rehydration trials (*p* = 0.001). At the end of rehydration, PV was significantly higher in the CEW trial compared to the CE trial (CE vs. CW vs: −0.77; 4.64, *p* = 0.001; 95% CI = −7.95, −2.88; ES = −0.78; CE vs. CEC: −0.77; 2.15, *p* = 0.020; 95% CI = −5.45, −0.39; ES = −0.56; CEW vs. CEC: 4.64; 2.15, *p* = 0.687; 95% CI = 0.049, 4.94; ES = 0.69). 

### 3.3. Subjective Feeling

Participants reported drinking CE (68 ± 12) to be sweeter than drinking CEW (27 ± 16) and CEC (17 ± 14; *p* = 0.030). Stomach upset was reported more in the CEC drink. Furthermore, no significant differences between trials were reported in subjective feelings of a mouthful (*p* = 0.575), thirst (*p* = 0.389), alertness (*p* = 0.136), bloatedness (*p* = 0.205), tiredness (*p* = 0.437), stomach fullness (*p* = 0.307), ability to concentrate (*p* = 0.146) or head feel (*p* = 0.355). 

### 3.4. Endurance Capacity Test

The mean endurance capacity test (2400 m running) time was significantly lower in CEW (11.4 ± 1.41 min) than in CEC [(14.25 ± 1.58 min); (*p* = 0.001)], but no significant differences were observed compared with CE [(12.9 ± 1.01 min); (*p* = 0.142); (Table 4)]. No significant differences were observed between CE and CEC trials (*p* = 0.215).

Mean skin temperature, heart rate, RPE, and thermal stress data obtained before and after the endurance capacity test are presented in Table 5. Mean skin temperature increased after the test but no differences were found between trials (*p* = 0.612). Perceived thermal stress (*p* = 0.018), RPE (*p* = 0.047), and heart rate (*p* = 0.042) were significantly higher in the CEC trial. 

## 4. Discussion 

The purpose of this study was to compare the rehydration capabilities of CE, CEW, and CEC drinks. The main finding of this study was that during 3 h of rehydration after 2.1 ± 0.2% body mass loss caused by a dehydration exercise: (1) ingesting CEW solution retained more fluid in the body than that of CE or CEC solution, and (2) ingesting of the CEW solution produced the least urine volume among all the three solutions, with urine loss of approximately 440 ML in the CEW trial compared with approximately 518 ML and 490 ML in the CE and CEC trials, respectively. Endurance capacity test time, heart rate, perceived thermal stress, and RPE in the CEW trial were also significantly lower than in other trials while the addition of casein caused stomach upset/distress and subsequent performance loss. To our knowledge, this study is the first to specifically investigate the rehydrating effect of two different proteins when added to a common CE solution on performance in warm, humid field conditions.

In a study by Judelson et al. (2007), it was demonstrated that hypohydration reduces muscle strength (2%), power (3%), and high-intensity endurance (10%) capacity [25]. According to a meta-analysis by Savoie et al. (2015), hypohydration decreased muscle strength by 5.5 ± 1.0% and decreased anaerobic power (−5.8 ± 2.3%) [26]. Hence, the process of rehydration assumes a crucial role in upholding cognitive function, concentration, and reaction time, all of which are imperative for optimal performance. Consequently, commencing training and engaging in competitive activities while in a state of optimal hydration and ensuring adequate fluid intake is essential for maximizing both performance and overall well-being [27]. Although several studies have been conducted to examine the need for fluid replenishment in controlled laboratory settings, there is a scarcity of research that investigates the physiological responses to exercise in warm and humid environments. Moreover, there is a paucity of knowledge regarding the effects of different protein types on rehydration status and exercise performance. 

The loss of >2% of body mass by sweating reduces exercise performance, muscular strength, and cognitive functions [25,28]. Consequently, several research studies have been undertaken to ascertain the expeditious replenishment of fluid loss subsequent to dehydration. The available evidence indicates that sports drinks that include CHO and electrolytes are generally more effective for rehydration purposes compared to plain water [29,30]. Nevertheless, recent research has shown that the use of protein-containing solutions for rehydration is more effective in preserving fluid balance compared to the ingestion of sports drinks [13,16,17,18]. Following this, there have been reports indicating that solutions containing protein have a greater impact on fluid retention in the body compared to CE [18,31]. Furthermore, comparable outcomes have been observed when comparing milk protein solutions with the CE solution. The present study demonstrated that the ingestion of a CEW solution resulted in greater fluid retention than a CEC or CE solution. Mean fluid retention was higher in the CEW trial (80.35 ± 5.9%) compared with the CE (76.67 ± 2.8%) and CEC trial (78.65 ± 2.6%). However, no significant difference was found between trials. Improved rehydration and fluid retention may be due to the presence of essential amino acids (AAs) [32]. Furthermore, it is well acknowledged that whey protein exhibits a more rapid digestion process and has a greater rate of absorption in comparison to casein [33]. Moreover, whey protein has been shown to be more efficacious in enhancing the levels of AAs in the bloodstream when compared to other protein sources, such as casein [34].

Studies demonstrated that AAs are effective for the absorption of sodium and water [35,36]. Hence, it may be anticipated that the absorption of AAs would lead to an increased uptake of salt, thereby causing a substantial osmotic/oncotic pressure inside the circulatory system. As a result, AAs have the potential to enhance the synthesis of plasma albumin, facilitating the absorption of more fluid into the circulation in order to maintain a consistent level of albumin [35]. Ultimately, a greater quantity of water has the capacity to be assimilated and maintained inside the circulatory system. However, different results were observed that there was no difference between the CE and whey protein trials for fluid retention [31]. Inconsistent results may be due to differences in protein amount and rehydration protocol. In a previous investigation, the quantity of whey protein was observed to be lower compared to the current study, with levels of 15 g/L as opposed to 22 g/L. It is plausible that the limited quantity of whey protein may not have yielded a statistically significant increase in fluid retention [31]. In the current investigation, in order to mitigate the occurrence of stomach bloating and the delay in gastric emptying, the participants consumed the solutions in six discrete portions at intervals of 30 min, resulting in a cumulative duration of 180 min. However, in another study, a much shorter rehydration time (60 min) was administered [31]. In contrast to whey protein, casein has a limited capacity to elevate blood AAs due to its comparatively slower digestion process inside the gastric environment [34]. Although AAs have been recognized for their role in facilitating the absorption and retention of fluid, the findings of this research indicate that casein did not provide any additional advantages in terms of fluid retention. At the conclusion of the rehydration process, the quantity of fluid retained was found to be equivalent in both the CE and CEC solutions.

The volume of urine produced during rehydration was similar in the CE and CEC trials, but more fluid was lost approximately 15% through urine in the CE than in the CEW trial. This outcome may be ascribed to the aforementioned characteristic of casein, namely its delayed digestion. Nevertheless, one may hypothesize that the presence of whey protein in the solutions resulted in an augmentation of fluid absorption and retention inside the body, as explained earlier. Consequently, this could have led to a greater degree of fluid retention compared to the CE or CEC solution. Further investigation is required to elucidate the underlying processes accountable for these findings.

Before the protocol, all participants had USG < 1.020 g/mL. This result shows that the participants were in a normal euhydrated state before the start of the test as previously described (3). USG significantly decreased at 2 h and 3 h time-points of rehydration in all three trials. However, compared to the CE and CEC, the CEW trial showed higher USG. Similar findings have been reported by previous research [16,37]. A reduction in USG may arise from increased urine output and subsequent dilution of urine [38]. In the present study, urine production was higher in CE and CEC trials than in CEW, which probably led to a decrease in USG.

Plasma volume (PV) was used to determine the fluid status of the body. PV decreases after exercise due to fluid movement from blood vessels to surrounding tissues and fluid loss through sweating [39]. After dehydration, an approximately 2.5% decrease of PV was observed. At the end of rehydration, PV was significantly higher in the CEW trial (4.64 ± 1.3%) and CEC (2.15 ± 1.3%) trial than in the CE (0.77 ± 2.7%). Similar to the results of the present research, in another study, it was noted that during a one-hour recovery time, the PV level exhibited an increase compared to the pre-exercise baseline, and this rise persisted until the conclusion of the recovery period [30].

The response of blood glucose in CE showed a greater increase during rehydration, which could be due to the high amount of carbohydrates in the CE (66 g/L) compared to the CEW (44 g/L) and CEC (44 g/L) trials. This result was predictable because previous studies demonstrated similar results after a solution with high CHO density was consumed by participants [40,41]. After a 3 h rehydration period, the glucose levels observed in the CE experiment did not exhibit a statistically significant increase compared to the other two trials. The results of the following physical capacity tests indicated a significant decrease in the CEW trial when compared to the other trials. 

To the best of our understanding, this research represents the first examination of the effects of including isolated whey and casein proteins in rehydration solutions after exercise, with a particular focus on their impact on performance in warm environmental conditions. The results showed a significant improvement in performance time in the CEW trial compared to the CE and CEC trials. In line with the current investigation, solitary research was undertaken that failed to document the advantageous impacts of rehydration using either low-fat milk or a CE sports beverage on exercise performance [15]. In contrast to the work previously described, the current study included milk proteins, namely isolated casein whey and casein, into a CE solution and then compared them individually. 

The evidence suggests that dehydration has a negative impact on exercise performance. In the current study, it was shown that a greater positive fluid balance was only detected in the CEW trial after a period of 3 h of rehydration. Additionally, it is worth noting that the blood glucose levels were consistent during the CEW trial, which may perhaps account for the observed enhancement in performance. The performance of those consuming the CEC beverage has been observed to decline as a result of experiencing gastrointestinal discomfort, albeit its superior rehydration capabilities. Further investigation is required to elucidate the underlying processes by which the consumption of CEW solution impacts physical performance during rehydration.

The findings derived from this research should be evaluated within the context of the following limitations. In the current investigation, participants were restricted from consuming fluids at their own discretion, ensuring that every participant drank an amount equivalent to 150% of their body mass loss. This approach was used to mitigate any potential differences in fluid consumption across participants. The measurement of plasma osmolality was not conducted in order to enhance the accuracy of assessing the rate of rehydration. Furthermore, it should be noted that the research did not use a crossover counterbalance design, which may provide potential confounding due to individual variations and may impact the findings. Finally, we acknowledge that our sample size may have been inadequate for detecting small differences in USG between different beverages, particularly considering the very small effect sizes based on previously reported literature [19]. Nonetheless, our sample size was considerably larger than other previously published studies in this research area [10,15,30,42] and future studies with larger recruitment samples would be beneficial to delineating more sensitive changes/differences in USG between different protein beverages.

In conclusion, our data demonstrate that the consumption of CEW solution subsequent to experiencing dehydration in a humid and hot environment leads to heightened fluid retention and plasma volume, resulting in reduced urine production throughout a 3 h recovery phase compared to the ingestion of CE or CEC solutions. The use of a CEW solution seems to exhibit more efficacy in facilitating rehydration compared to a CE or CEC solution during a limited duration of recovery after exercise-induced dehydration. Moreover, the CEW solution resulted in lower heart rate and RPE than other solutions, as well as improved physical capacity. While CEC had a positive impact on fluid retention rates, participants exhibited suboptimal performance as a result of experiencing stomach discomfort. 

## Figures and Tables

**Figure 1 nutrients-15-04374-f001:**
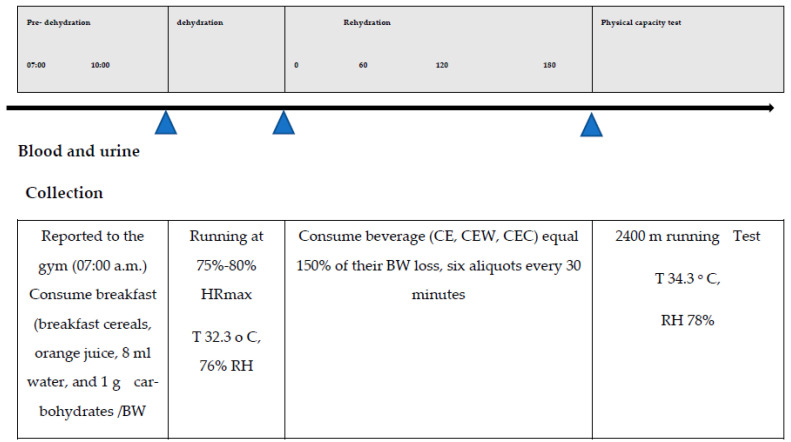
The experimental design of the study.

**Figure 2 nutrients-15-04374-f002:**
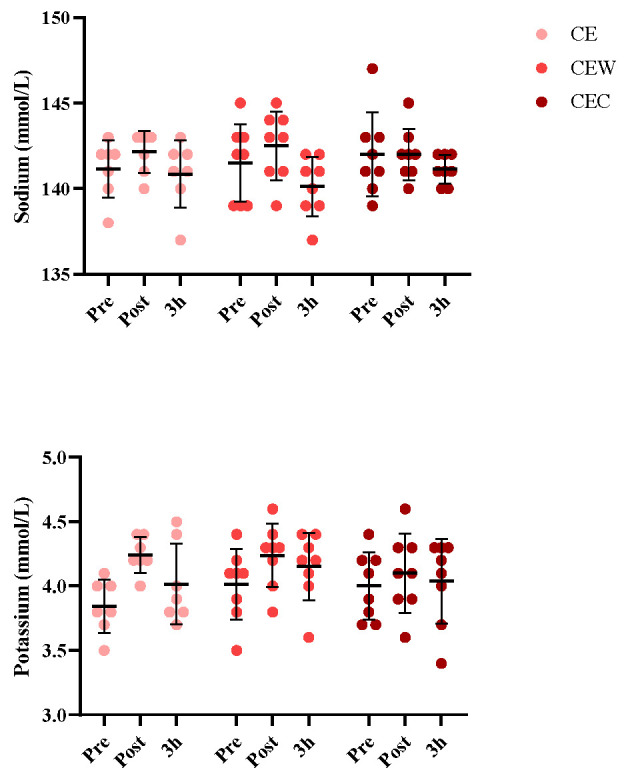
Serum and urine levels of sodium and potassium. Error bars indicate standard deviation. CE, carbohydrate-electrolyte solution; CEW, carbohydrate-electrolyte-whey protein solution, CEC, carbohydrate-electrolyte-casein protein solution.

**Figure 3 nutrients-15-04374-f003:**
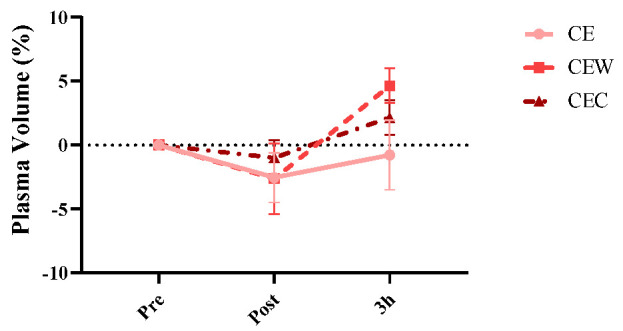
Percent change in plasma volume. Error bars indicate standard deviation. CE, carbohydrate-electrolyte solution; CEW, carbohydrate-electrolyte-whey protein solution, CEC, carbohydrate-electrolyte-casein protein solution.

**Table 1 nutrients-15-04374-t001:** Composition of rehydration solutions.

	CE	CEW	CEC
Carbohydrate (g)	66	44	44
Protein (g)	0	22	22
Sodium (mM)	14	14	14
Potassium (mM)	3. 3	3. 3	3. 3
Energy (kcal/L)	264	264	264

Abbreviations. CE, carbohydrate-electrolyte solution; CEW, carbohydrate-electrolyte-whey protein solution, CEC, carbohydrate-electrolyte-casein protein solution.

**Table 2 nutrients-15-04374-t002:** Baseline characteristics of the participants.

Variable	CE	CEW	CEC
Measure			
Anthropometry			
Age (y)	24.3 ± 2.4	24.6 ± 2.1	24.2 ± 1.8
Body mass (kg)	75.2 ± 3.6	74.7 ± 4.1	76.5 ± 5.2
Stature (cm)	176.4 ± 6.8	177.1 ± 5.4	176.6 ± 7.3
BMI (kg.m^−2^)	24.2 ± 1.5	23.9 ± 1.3	24.3 ± 1.7
VO_2max (_mL/kg/min_)_	49.8 ± 4.1	48.5 ± 3.9	50.2 ± 4.4
Biochemical markers			
USG (g/mL)	1.019 ± 0.002	1.018 ± 0.002	1.018 ± 0.002
Glucose (mg/dl)	91.33 ± 6.26	93.25 ± 7.41	91.68 ± 5.17
Na (mmol/L)	141 ± 1.67	141± 2.1	141 ± 1.49
K (mmol/L)	3.8 ± 0.2	4 ± 0.2	4 ± 0.1
Hemoglobin (g/dL)	14.47 ± 0.71	14.95 ± 0.62	14.96 ± 0.85
Hematocrit (%)	44.02 ± 1.93	43.72 ± 1.42	44.51 ± 1.14
PV (ml)	2883.42 ± 201.79	2800.25 ± 216.54	2852 ± 231.76

Abbreviations. BMI, body mass index; VO_2max_, maximal oxygen consumption; USG, urine specific gravity, NA, sodium; K, potassium; PV, plasma volume; CE, carbohydrate-electrolyte solution; CEW, carbohydrate-electrolyte-whey protein solution, CEC, carbohydrate-electrolyte-casein protein solution.

**Table 3 nutrients-15-04374-t003:** Body mass loss, dehydration time, and fluid volume consumed within trials.

Variable	Trial	Mean ± SD	ANOVA *p*-Value
Body mass loss (kg)	CE CEW CEC	1.61 ± 0.17 1.62 ± 0.11 1.61 ± 0.12	*p* = 0.997
Dehydration time (min)	CE CEW CEC	11.42 ± 2.17 12.75 ± 1.53 12.63 ± 1.27	*p* = 0.418
**Volume of fluid consumed during the rehydration period (mL)**	**CE** **CEW** **CEC**	**2235.71 ± 188.66** **2287.5 ± 203.10** **2303.75 ± 246.05**	***p* = 0.820**

**Abbreviations. CE, carbohydrate**-electrolyte solution; CEW, carbohydrate-electrolyte-whey protein solution, CEC, carbohydrate-electrolyte-casein protein solution.

**Table 4 nutrients-15-04374-t004:** Effect of rehydration beverage on physical capacity.

Variable	Trial	Mean ± SD	ANOVA	Pairwise Comparison	ES (95% CI)
Physical Capacity	CE CEW CEC	12.9 ± 1.01 11.4 ± 1.41 14.25 ± 1.58	*p* = 0.002	CE vs CEW: *p* = 0.142 CE vs CEC: *p* = 0.215 CEW vs CEC: *p* = 0.001	0.7 (−0.35, 3.35) 0.29 (−3.2, 0.50) −0.59 (−4.64, −1.05)

**Abbreviations.** CE (carbohydrate-electrolyte), CEW (CE plus whey protein), CEC (CE plus casein protein) drink. ES: effect size; 95% CI: 95% confidence interval.

**Table 5 nutrients-15-04374-t005:** Mean skin temperature, heart rate, and ratings of perceived exertion and thermal stress in pre and endurance capacity test.

	Pre	Post
**Mean Skin Temperature**		
**CE**	37.7 ± 0.3	37.3 ± 0.5
**CEW**	37.2 ± 0.7	37.5 ± 0.6
**CEC**	37.5 ± 0.8	37.9 ± 0.8
**Heart Rate**		
**CE**	75 ± 8	181 ± 6
**CEW**	76 ± 2	178 ± 4
**CEC**	78 ± 5	187 ± 4
**Ratings of Perceived Exertion**		
**CE**	-	17 ± 2
**CEW**	-	16 ± 2
**CEC**	-	19 ± 2
**Thermal Stress (10 unbearable cold; 10 unbearable heat)**		
**CE**	3 ± 1	7 ± 1
**CEW**	3 ± 1	7 ± 2
**CEC**	3 ± 1	9 ± 2

**Abbreviations.** CE (carbohydrate-electrolyte), CEW (CE plus whey protein), CEC (CE plus casein protein) drink.

## Data Availability

Data sharing is applicable.

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
