# Peer review of "Isolate Whey Protein Promotes Fluid Balance and Endurance Capacity Better Than Isolate Casein and Carbohydrate-Electrolyte Solution in a Warm, Humid Environment"

_nutrients, 2023, doi:10.3390/nu15204374_

Round 1

Reviewer 1 Report

The authors completed a randomized controlled trial to investigate the effects of whey isolate or casein on the restoration of body fluid-electrolyte equilibrium, regulation of body temperature, and subsequent physical performance during exercise in warm and humid conditions among a group of young male soldiers. This is a well-organized and well-presented study. I have only two comments:

1. Power calculation is essential for this study, given that there are previous similar studies.  Please provide a post hoc power calculation and put it in perspective with the outcomes.

2. Please report the randomization method

Author Response

Dear Reviewer

Thank you very much for giving us the opportunity to respond to the reviewers’ comments and revise our manuscript according to the journal’s standards. We are pleased to clarify your concerns, which we believe will improve the impact and quality of our work. Please find below our responses to your observations. We have made a concerted attempt to systematically address the specific concerns raised for this revision.

The authors completed a randomized controlled trial to investigate the effects of whey isolate or casein on the restoration of body fluid-electrolyte equilibrium, regulation of body temperature, and subsequent physical performance during exercise in warm and humid conditions among a group of young male soldiers. This is a well-organized and well-presented study. I have only two comments:

Authors: We appreciate your kind feedback on our manuscript.

  1. Power calculation is essential for this study, given that there are previous similar studies.  Please provide a post hoc power calculation and put it in perspective with the outcomes.

Authors: We thank the reviewer for raising this important point regarding sample size. During the planning/concept phase of our work, we considered performing power calculations for our primary outcome measure of urine specific gravity (USG). Most studies in the literature report USG in figures, precluding the capacity to use mean and standard deviations for our calculations. One study by Li and colleagues (PMID: 29490577) that did report mean and standard deviation measures for USG between whey and soy beverages in a table provided an effect size of only 0.02; when putting this information into G-power, a sample size of over 100 would have been required to be recruited for our work. As this is logistically impossible for multiple reasons, we recruited a participant cohort size that is well above other previously published studies in this research area (example: PMIDs: 29696654, 18618137, 21813913, and 18463891). We understand and appreciate that this approach does not preclude the capacity to observe Type 1 error in our work (PMID: 27375390), we have mentioned this is an important limitation in our findings and thus believe we have transparently addressed this consideration.

  1. Please report the randomization method

Authors: Participants were randomly assigned to one of three groups using an online resource (www.randomizer.org). This has been added to the methods section.

Reviewer 2 Report

The authors investigated the effect of the addition of milk-derived whey or casein protein to carbohydrate-electrolyte drinks on post-exercise rehydration and endurance capacity. Based on the results, they concluded that the inclusion of isolate whey protein in a carbohydrate-electrolyte solution was superior compared to consuming the carbohydrate-electrolyte solution or in conjunction with isolate casein protein. The study is useful for some people who are working in some special circumstances. Some comments or issues are listed below:

1) How many times were the experiments performed? They should indicated in the manuscript.

2) Figure 3 could be changed to the histogram.

3) All the soldiers did experiments indoors?

4) Replace the collection figure in Figure 1 with text.

It is fine to read and understand.

Author Response

Dear Reviewer

Thank you very much for giving us the opportunity to respond to the reviewers’ comments and revise our manuscript according to the journal’s standards. We are pleased to clarify your concerns, which we believe will improve the impact and quality of our work. Please find below our responses to your observations. We have made a concerted attempt to systematically address the specific concerns raised for this revision.

The authors investigated the effect of the addition of milk-derived whey or casein protein to carbohydrate-electrolyte drinks on post-exercise rehydration and endurance capacity. Based on the results, they concluded that the inclusion of isolate whey protein in a carbohydrate-electrolyte solution was superior compared to consuming the carbohydrate-electrolyte solution or in conjunction with isolate casein protein. The study is useful for some people who are working in some special circumstances. Some comments or issues are listed below:

1) How many times were the experiments performed? They should indicated in the manuscript.

Authors: All trials were conducted once as conveyed in methods section.

2) Figure 3 could be changed to the histogram.

Authors: We thank the reviewer for this insightful and valid suggestion. We have actually constructed this figure in accordance with previously published research on this subject matter (PMIDs: 18618137. 21813913, 18463891) thus believe this figure to be appropriate in its current form.

3) All the soldiers did experiments indoors?

Authors: That’s correct; all soldiers completed experiments indoors.

4) Replace the collection figure in Figure 1 with text.

Authors: Thanks; this has now been completed.
